# Chemoradiotherapy Combined with Brachytherapy for the Definitive Treatment of Esophageal Carcinoma

**DOI:** 10.3390/cancers15143594

**Published:** 2023-07-12

**Authors:** Julian Mangesius, Katharina Hörmandinger, Robert Jäger, Sergej Skvortsov, Marlen Plankensteiner, Martin Maffei, Thomas Seppi, Daniel Dejaco, Matthias Santer, Manuel Sarcletti, Ute Ganswindt

**Affiliations:** 1Department of Radiation Oncology, Medical University of Innsbruck, 6020 Innsbruck, Austria; 2Comprehensive Cancer Center Innsbruck (CCCI), 6020 Innsbruck, Austria; 3Department of Radiotherapy, State Hospital of Bolzano, 39100 Bolzano, Italy; 4Department of Otorhinolaryngology-Head and Neck Surgery, Medical University of Innsbruck, 6020 Innsbruck, Austria

**Keywords:** esophageal cancer, brachytherapy, dose escalation, chemoradiotherapy

## Abstract

**Simple Summary:**

The aim of this retrospective study was to investigate whether dose escalation with additional intraluminal brachytherapy after definitive chemoradiotherapy can improve local control and survival in esophageal cancer. In the 71 patients who received additional brachytherapy, median local progression-free survival, as well as overall survival, significantly improved in comparison to the 112 patients who received chemoradiotherapy alone. There was an increase in late toxicity with the addition of brachytherapy; however, no difference in life-threatening or lethal toxicities was observed. This concept can be a safe and effective option for dose escalation in chemoradiotherapy regarding esophageal cancer.

**Abstract:**

This study aims to investigate the effect of dose escalation with brachytherapy (BT) as an addition to definitive chemoradiotherapy (CRT) on local control and survival in esophageal cancer. From 2001 to 2020, 183 patients with locally limited or locally advanced esophageal cancer received definitive CRT with or without brachytherapy in a two-center study. External-beam radiotherapy was delivered at 50.4 Gy in 1.8 Gy daily fractions, followed by a sequential boost to the primary tumor of 9 Gy in 1.8 Gy daily fractions if indicated. Intraluminal high dose rate (HDR) Ir-192 brachytherapy was performed on 71 patients at 10 Gy in two fractions, with one fraction per week. The combined systemic therapy schedules used included 5-fluorouracil/cisplatin or 5-fluorouracil alone. Cisplatin was not administered in patients receiving brachytherapy. The median local progression-free survival was significantly extended in the BT group (18.7 vs. 6.0 months; *p* < 0.0001), and the median local control was also significantly prolonged (30.5 vs. 11.3 months, *p* = 0.008). Overall survival (OS) significantly increased in the BT group (median OS 22.7 vs. 9.1 months, *p* < 0.0001). No significant difference in the overall rate of acute toxicities was observed; however, the rate of acute esophagitis was significantly higher in the BT group (94.4% vs. 81.2%). Likewise, the overall rate of late toxicities (43.7% vs. 18.8%) was significantly higher in the BT group, including the rate of esophageal stenosis (22.5% vs. 9.8%). There was no difference in the occurrence of life-threatening or lethal late toxicities (grades 4 and 5). Brachytherapy, after chemoradiation with single-agent 5-FU, represents a safe and effective alternative for dose escalation in the definitive treatment of esophageal cancer.

## 1. Introduction

Esophageal cancer is the sixth leading cause of cancer-related mortality worldwide. Most patients present with locally advanced tumors, with a poor overall survival (OS) rate of only 15–25% after 5 years. In patients treated with definitive chemoradiotherapy (CRT), high rates of locoregional failure are a concern and have been observed in several prospective investigations [1,2,3,4,5]. The RTOG 85-01 study [1] compared CRT, including 5-fluorouracil (5-FU) and cisplatin (with 50 Gy delivered to the tumor and 30 Gy to the esophagus, including regional lymph nodes) to radiotherapy (RT) alone (with 64 Gy). Despite improved survival for CRT, the rate of local and locoregional failure, and the rate of persistent tumor together remained high at 52%, with the primary pattern of failure being persistent disease (26%). Patterns of failure analyses have consistently shown local failure at the primary tumor site to be the main factor causing treatment failure [3,5]. Two randomized trials have sought to improve local control (LC) with dose escalation [2,6]. The INT 0123 trial [2] investigated whether adding a sequential boost of 14.4 Gy to the tumor in addition to 50.4 Gy to the primary and regional lymph nodes in CRT with 5-FU and cisplatin would increase OS in comparison to standard CRT with 50.4 Gy. No significant benefit in either OS (13.0 vs. 18.1 months) or locoregional control (56% vs. 52%) was found for high-dose CRT. 

The ARTDECO study [6] revisited the concept of dose escalation in a modernized trial investigating CRT with carboplatin plus paclitaxel and 50.4 Gy/1.8 Gy given to the tumor and regional lymph nodes, as well as a simultaneous integrated boost (SIB) of 61.6 Gy to the primary lesion, and this was compared to CRT without the boost. Despite incorporating advances in therapy, when compared to previous CRT trials, by using a better-tolerated chemotherapy regimen (instead of 5-FU/cisplatin), using modern intensity-modulated (IMRT) treatment delivery, and integrating the boost dose (thus increasing the biologically effective dose), still no significant improvement in local progression-free survival (LPFS) could be achieved (3-year LPFS 73% vs. 70%). 

Dose escalation using brachytherapy instead of external beam radiotherapy (EBRT) has been investigated in the RTOG 92-07 phase I/II trials [7]. High dose rate (HDR) brachytherapy with 15 Gy or 10 Gy (three or two times 5 Gy at 10 mm from the mid-source position) was delivered after EBRT with 50 Gy, combined with concurrent 5-FU and cisplatin. Treatment toxicity was a concern, and the HDR dose was reduced from three to two applications after multiple occurrences of esophageal fistulas (12%). Locoregional failure occurred in 23% of patients and persistent disease in 19%. The rate of failure at the primary site in this single group trial was, thus, lower in comparison to the CRT group of the RTOG 85-01 trial; however, this was at the cost of unacceptable toxicity. 

Despite a clear rationale for dose escalation to improve locoregional control, as well as evidence for improved control with higher doses in nonrandomized trials [8], two randomized controlled trials have not proven any benefit so far. While treatment toxicity was suspected to play a role, no clear cause for the lack of success in dose escalation has been established. 

This study aimed to investigate whether dose escalation with brachytherapy, in addition to definitive CRT, can improve LPFS.

## 2. Materials and Methods

This is a retrospective two-center study of patients receiving definitive CRT with or without brachytherapy for esophageal cancer. Consecutive patients treated at the general hospital of Bolzano, Italy (N = 66), and at the university clinics of Innsbruck, Austria (N = 112), were included from 2001 to 2020. Patients receiving neoadjuvant CRT or palliative brachytherapy alone without EBRT were excluded. The patient and treatment characteristics of the 183 included patients are displayed in Table 1. The study was approved by the ethics commission of the Medical University of Innsbruck.

All patients received EBRT to the primary and regional nodes (including the paraoesophageal nodes with or without the coeliac and paraclavicular nodes) with 50.4 Gy in 1.8 Gy daily fractions at five fractions per week for 5.5 weeks. A sequential EBRT boost to the primary tumor of 9 Gy in 1.8 Gy daily fractions was performed if indicated (76.5% of all patients; N = 140). Specifically, those patients in which single-agent 5-FU was used in combined chemoradiotherapy were eligible for an EBRT boost. The rationale of this concept was to avoid the excessive toxicity expected when using combined 5-FU and cisplatin chemoradiotherapy while avoiding a loss in tumor control due to systemic undertreatment, especially in patients receiving brachytherapy. In 82.8% of these, the boost could be performed by adhering to institutional dose constraints. In all other patients, the application of an EBRT boost was performed at the discretion of the treating physician. 

EBRT was delivered through 3D-conformal RT in the majority of patients (88.5%) or with intensity-modulated RT (11.5%). Brachytherapy, in addition to EBRT, was performed exclusively in Innsbruck, whereas the patients in Bolzano received EBRT only. Patients were not eligible for BT if the treated length exceeded 10 cm in case of stenosis that was unpassable by the applicator, the suspected presence of fistula, tumors located in the proximal cervical esophagus, or in case of the involvement of the gastroesophageal junction or cardia [9]. Patients had to be suitable for anesthesia and provide informed consent. None of the patients received endoscopic laser debulking prior to brachytherapy. 

Intraluminal brachytherapy was performed as 2D, planned HDR, after loading with Ir-192 with 10 Gy prescribed at a mucosal tissue depth of 5 mm in two fractions (5 Gy each), one fraction per week, starting two weeks after the completion of EBRT (in accordance to the GEC ESTRO Handbook of Brachytherapy [10]). The maximum applicator surface dose was limited to 11 Gy. The length of the planning target volume included the extent of the tumor plus 2 cm proximal and distal to the lesion. Dose planning was performed using Oncentra Brachy (Elekta Solutions AB, Stockholm, Sweden) with AAPM TG-43 formalism. The reference dose was calculated for a mucosal depth of 5 mm. The proximal and distal end of the tumor was marked using radio-opaque surgical clips that were placed during esophagoscopy a day before the first brachytherapy session. Positioning of the applicator (Bonvoisin-Gérard Esophageal Applicator Set, Nucletron B.V., Veenendaal, NL) was performed using a mobile X-ray device (c-arm) for visualization. A mouthpiece was used for the fixation of the positioned applicator. The applicator diameter ranged from 8–12 mm (median 10), using the largest possible applicator to achieve a flatter dose gradient and minimize mucosal surface dose. 

The combined systemic therapy schedules used included 5-fluorouracil 350 mg/m^2^/d continuously for 5 days per week, d1-22 (until 39.6 Gy reached); cisplatin 75 mg/m^2^ at d1 and d29 and 5-FU 1000 mg/m^2^ continuously over 96 h at d1 and d29. Platinum-based therapy was not administered in patients receiving brachytherapy. 

Treatment outcome (local and distant tumor control) was assessed by esophagoscopy and CT follow-up every 3 months for the first year after treatment, every 6 months for the second and third years, and yearly thereafter. The primary endpoint of the study was LPFS. The secondary endpoints included LC and OS. 

Statistical analysis was performed using R (V4.0, R Core Team, Vienna, Austria) with ‘Survival’ and ‘Survminer’ packages, as well as SPSS Statistics (V26, IBM Cooperation, Armonk, NY, USA). Patient characteristics were compared using a two-sided chi-squared test or independent sample *t*-tests for the continuous variables. For the survival analysis of LPFS, OS, and LC, the Kaplan–Meier method was used with a log-rank test for comparisons between the treatment groups. The Cox proportional hazard model was used for the uni- and multivariate analyses of the factors associated with LPFS, OS, and LC. Multivariate analysis was performed by applying the rule of stepwise backward eliminations of nonsignificant factors. Differences in the incidence of the individual and summarized acute and late toxicities were assessed by using the chi-squared test. 

In order to equalize the prognostic factors among the two compared patient cohorts, propensity matching (1:1 matching without replacement) was performed. Propensity matching yielded 68 comparable patient pairs with similar distributions of key prognostic factors (Appendix A). Propensity scores were calculated using multivariable logistic regression with the covariates age, KPS, UICC stage, histologic grade, and tumor location. Furthermore, those patients not receiving chemotherapy were excluded from the propensity-matched dataset. LPFS, OS, and LC were compared using the Kaplan–Maier method and the log-rank test in the matched pair sample. 

## 3. Results

### 3.1. Outcome

Median LPFS was significantly longer in the BT (18.7 months) vs. the no BT group (6.0 months; *p* < 0.0001, Figure 1c). The univariate analysis of the predictive factors for LPFS showed brachytherapy (no BT vs. BT; HR = 2.03, *p* <0.0001), tumor stage, and tumor grade significantly affected LPFS. For the multivariate analysis, no brachytherapy (HR = 1.66, *p* = 0.011) and tumor stage were confirmed to independently impact LPFS (Figure 2). The hazard ratio for LPFS was 0.49 for brachytherapy when compared to the controls. The benefits from brachytherapy were demonstrated for all patient subgroups, except for those under-represented in the study cohort (female patients, no combined chemotherapy) and for tumors located in the middle esophagus (HR 0.76, CI 0.45–1.30, Figure A2).

Median LC in the BT group was significantly better (30.5 vs. 11.3 months, *p* = 0.008; Figure 1b). For the single variable analysis of the predictive factors, only brachytherapy (no BT vs. BT; HR = 2.00, *p* = 0.009) and tumor stage (IVa vs. I; HR = 9.28, *p* = 0.035) were associated with LC (Table 2). No multivariate analysis was performed. LC, after 12 months, was 80.6% with brachytherapy and 46.4% without.

There was no significant difference in the distant controls (DCs) between the treatment groups (no BT: median 92.5 months; with BT: 84.3 months, *p* = 0.214, Figure A1).

OS was significantly prolonged in the BT group (median OS 22.7 vs. 9.1 months, *p* < 0.0001, Figure 1a). The univariate analysis of the predictive factors for OS showed that brachytherapy (no BT vs. BT; HR= 2.04, *p* < 0.001) and tumor stage significantly affected OS. For the multivariate analysis, brachytherapy (no BT HR = 1.74, *p* = 0.004) and tumor stage (IVb vs. I; HR 2.90, *p* = 0.036) also significantly impacted OS (Figure 3).

In the propensity-score-matched population, median LPFS remained significantly improved in the BT group (15.7 vs. 6.0 months, *p* < 0.001). Likewise, the same was valid for OS (18.7 vs. 8.2 months, *p* = 0.006) and LC (30.4 vs. 10.9 months, *p* = 0.019). 

The propensity score analysis of center-specific OS and LPFS was performed after excluding the BT patients and showed no statistically significant difference in median OS (Bolzano 9.8 months vs. Innsbruck 6,2 months, *p* = 0.863, Appendix A) and median LPFS (Bolzano 9.6 months vs. Innsbruck 9.11 months, *p* = 0.702, Appendix A).

### 3.2. Treatment-Related Toxicity

Overall, the rates of toxicity were comparable between the groups. The summarized analysis of the accumulative acute toxicities showed no difference in the rate of toxicities between the two treatment groups (Table 3, Figure A3). However, an analysis of the single acute toxicity items revealed that the rate of the grade (1–3) of acute esophagitis was significantly higher in the BT group (94.4% vs. 81.2%; *p* < 0.05; Table A1). Grade 3 esophagitis occurred in 9.9% of patients with BT and in 8.0% of patients without BT. No esophagitis > Grade 3 was observed. Death was attributed to treatment-related acute toxicity in three patients (two patients who did not receive BT died of esophageal perforation, and one patient in the BT group died of acute pneumonitis; see Table 3 and Table A1). 

The overall rate of lower-grade (1–3) late toxicities was significantly higher in the BT group (Table 4, Figure A4). Likewise, the rate of esophageal stenosis was significantly higher in the BT group (Grade 1–3 22.5% vs. 9.8%; *p* < 0.05; Table A2). No Grade 4 or 5 stenosis was detected. No significant difference in other late toxicities (late pneumonitis, fistula, perforation, or bleeding) was detected between the treatment groups.

Late toxicities resulting in death occurred in five patients. One patient without BT died from an esophageal fistula. Esophageal bleeding led to the death of three patients in the BT group and of one patient in the group without BT. 

Overall, death was attributed to treatment-related toxicity in eight patients (four with and four without BT).

The application of both an EBRT boost and brachytherapy in 63 patients did not lead to a significantly elevated acute or late toxicity in comparison to those who received only either an EBRT boost or brachytherapy (see Appendix A). 

## 4. Discussion

In the current treatment concepts of esophageal cancer, the role of brachytherapy is often limited to palliative applications for local symptom alleviation. In contrast, this study evaluates the potential of additive intraluminal brachytherapy in the curative setting. For this purpose, we investigated the benefit of adding brachytherapy to standard CRT in the primary, curative treatment of esophageal cancer. We found that the use of BT was associated with significantly longer LPFS and LC when compared to standard CRT alone. Specifically, the median LPFS was 18.7 months in the BT group vs. 6.0 months without BT, while the median LC was 30.5 vs. 11.3 months. These results suggest that BT is an effective treatment option for improving LC in locally advanced esophageal cancer. Furthermore, while we observed slightly elevated rates of acute esophagitis and late esophageal stenosis, the overall toxicity was similar between the two groups. Late Grade 4 or 5 (life-threatening or leading to death) toxicities were observed in 5.6% of patients receiving BT vs. 2.7% without. The severe or fatal toxicities included bleeding (four vs. two patients), fistulas (zero vs. one patient), and perforation (one vs. four patients). Importantly, the occurrence of esophageal fistulas in the BT group was comparatively low and not significantly elevated (7% vs. 5.4%).

Our findings of improved LC are corroborated by other trials incorporating BT in definitive treatment concepts for esophageal cancer, most of which are discussed in two recent large review publications [11,12]. Several retrospective series have investigated the addition of brachytherapy for superficial tumors [13,14,15,16,17,18,19,20,21], as well as locally advanced tumors with [22,23,24,25,26,27] or without chemotherapy [28,29,30,31,32,33,34]. The addition of brachytherapy to CRT in the RTOG 9207 study led to an improvement in patients achieving complete remission in comparison to patients treated with CRT alone (RTOG 8501 study): from 68% to 74%. The rate of locoregional failure also decreased from 29% to 18%. However, the absolute rate of treatment failures was similar in both study cohorts, and OS did not improve (1 year: 49% vs. 57%; 3 years: 29% vs. 27%). It stands to reason that any potential improvement in survival through improved LC was compromised by excessive toxicity. The trial was criticized for the high BT dose of 15 Gy in three fractions with concurrent chemotherapy during both EBRT and BT. In a recent trial, 64 patients were treated with EBRT (50 Gy) and HDR brachytherapy (2 x 10 Gy) or EBRT alone (60 Gy). Both LC (2-year 53% vs. 22%) and OS (3-year 38% vs. 9%) markedly improved in the BT arm [35]. In a small series of 21 patients treated with EBRT (50–56 Gy) and an HDR brachytherapy boost of 3 x 4 Gy, when compared to EBRT alone (60–69 Gy), Nishimura et al. [36] found an improved 3-year LC of 85% vs. 45%, as well as better disease-specific survival. There was no significantly elevated toxicity with the addition of brachytherapy. Similarly, a trial including 50 patients receiving EBRT (35 Gy) without chemotherapy compared an HDR BT boost of 12 Gy in two fractions to an EBRT boost of 20 Gy in 10 fractions. After one year, LC was 70.6% in the BT group and 25.0% in the control group, and the relief of dysphagia was also better with BT [37]. Additionally, Sharan et al. [38] reported high rates of overall LC of 61.5% in a small, single-arm cohort of 26 patients treated with CRT (50.4 Gy) followed by HDR BT at 2 x 4 Gy. One patient in that series died of a tracheoesophageal fistula. In a series of 14 patients treated for primary or recurrent esophageal cancer, Folkert et al. [39] found that using a larger diameter applicator of 12–14 mm could reduce treatment toxicity by lowering the mucosal dose and ensuring more homogenous dose delivery. 

It is noteworthy that local failure at the primary tumor site and persistent primary tumors remain the most common patterns regarding treatment failure after CRT, indicating the need to improve LC to reduce cancer-related mortality. Treatment failure within the GTV was also shown to significantly impair OS. [5]

However, previous attempts at dose escalation using an EBRT boost have been unsuccessful in randomized trials, despite the indications of potential benefits regarding dose escalation in multiple smaller series [8,40,41,42]. While the reasons for the lack of proven benefits from EBRT dose escalation for LC and OS are still a matter of discussion, one likely explanation is that the application of a potentially more effective boost dose to reach >66 Gy within the GTV is limited by toxicity. Alternatively, the addition of brachytherapy to CRT with 5FU and cisplatin was shown to improve LC in the RTOG 92-07 trial [7]; however, unacceptable toxicity and especially the escalating risk of fistulas and ulcers have prevented the adoption of this promising technique in clinical practice so far. The main benefit of adding brachytherapy in comparison to an EBRT boost is the improved dose distribution, with substantially higher possible dose deliveries at the surface, accompanied by a steep dose gradient, thus offering a better therapeutic ratio. The higher dose per application also provides an elevated biologically effective dose at the tumor site. In order to address the toxicity observed in previous trials that combined CRT with brachytherapy, we opted to omit cisplatin-based chemotherapy for patients receiving brachytherapy and instead administered 5-FU 350 mg/m^2^/d continuously for 5 days per week during the first 22 days of the treatment. The de-escalation of the combined chemotherapy did not lead to significantly impaired distant control (distant control: 84.3 months with BT vs. 92.5 months). Furthermore, we performed brachytherapy in two fractions after the conclusion of CRT, each application 1 week apart at a reduced dose of 5 Gy per fraction measured to a tissue depth of 5 mm. We also aimed to limit the dose at the esophageal surface to 11 Gy per session. When taken together, these measures led to a more favorable toxicity profile in comparison to the RTOG 92-07 trial. 

The patients investigated in this two-center study were not randomized. The patients treated in the Italian center received state-of-the-art definitive CRT with an identical therapy concept (regarding dose prescription, EBRT boost, chemotherapy schedules, and follow-up) to the Austrian center. The option of brachytherapy dose escalation was only available at the Austrian center, and it was applied whenever indicated and technically feasible. Despite the lack of randomization, the difference in availability of brachytherapy with otherwise identical treatment concepts at the two geographically neighboring centers ensured the comparability of the treatment groups. We cannot exclude a selection bias due to the retrospective nature of the study, resulting in some unavoidable imbalances in the distribution of favorable and unfavorable prognostic factors between the treatment groups (KPS, staging, location, and chemotherapy). Moreover, platinum-based chemotherapy was also under-represented in the control group, which may, in part, explain the large observed difference in OS. The study also included a small number of patients with stage IVb (with limited M1) esophageal cancer, who, nonetheless, received chemoradiotherapy and, in seven cases, also received brachytherapy. Recent data collected by Matoska et al. also show a survival benefit with definitive CRT in oligometastatic patients [43]. Although we would generally not recommend definitive EBRT and BT for these patients, we also observed a favorable (however not significant) median LPFS (median 9.2 vs. 3.1) and OS (18.7 vs. 4.4 months) in this small, advanced-stage subgroup.

In order to increase the robustness of the statistical analysis and thereby support the validity of our primary results, a multivariate analysis was performed to account for the mentioned potential confounding factors. Finally, to further minimize any bias in patient selection potentially distorting our results, a propensity-score-matched pair analysis was performed. Therefore, an accurate estimate of the LPFS, LC, and OS times could be confirmed in the balanced cohort.

## 5. Conclusions

When taken together, the demonstrated feasibility and favorable locoregional control observed in this study provide evidence to warrant revisiting the concept of CRT with brachytherapy in future randomized controlled studies. We demonstrated that excessive toxicity in the definite treatment of esophageal cancer patients could be avoided by modifying the boost treatment concept and omitting cisplatin in combined chemotherapy. This concept presents a treatment alternative, especially for patients unsuited for surgery or cisplatin-based combined chemotherapy.

## Figures and Tables

**Figure 1 cancers-15-03594-f001:**
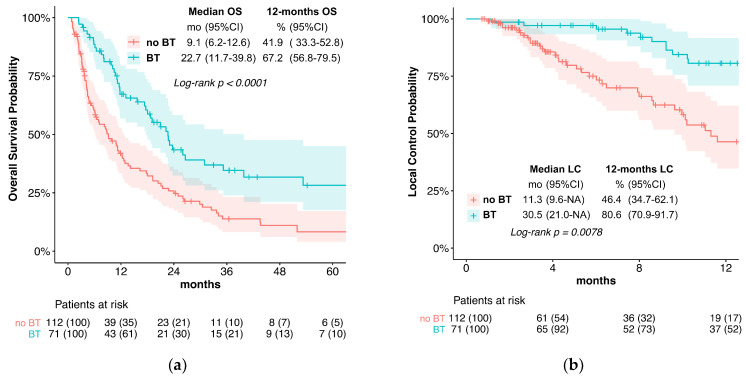
(**a**) Overall survival; (**b)** local control; (**c**) local progression-free survival.

**Figure 2 cancers-15-03594-f002:**
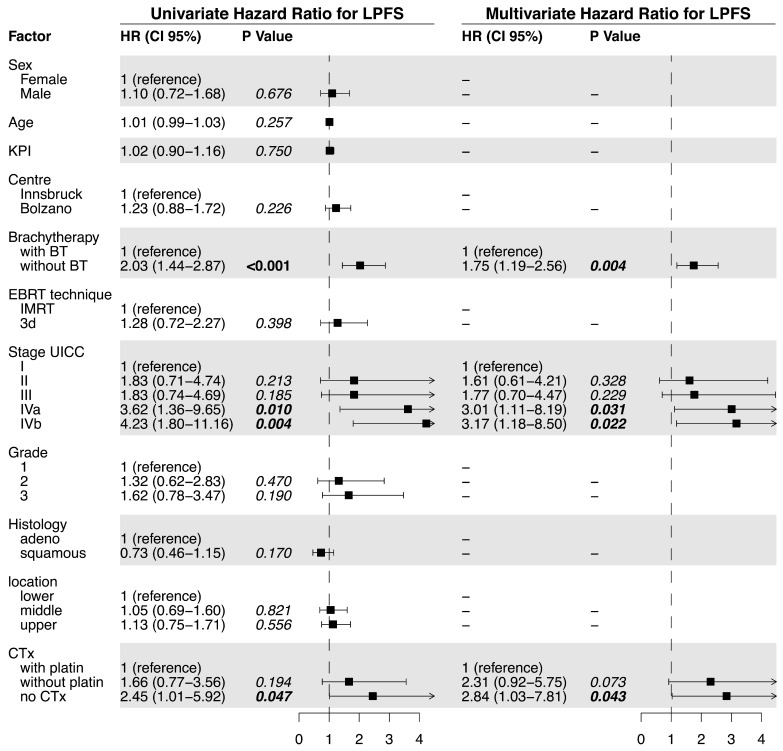
Univariate and multivariate analysis of factors related to local progression-free survival.

**Figure 3 cancers-15-03594-f003:**
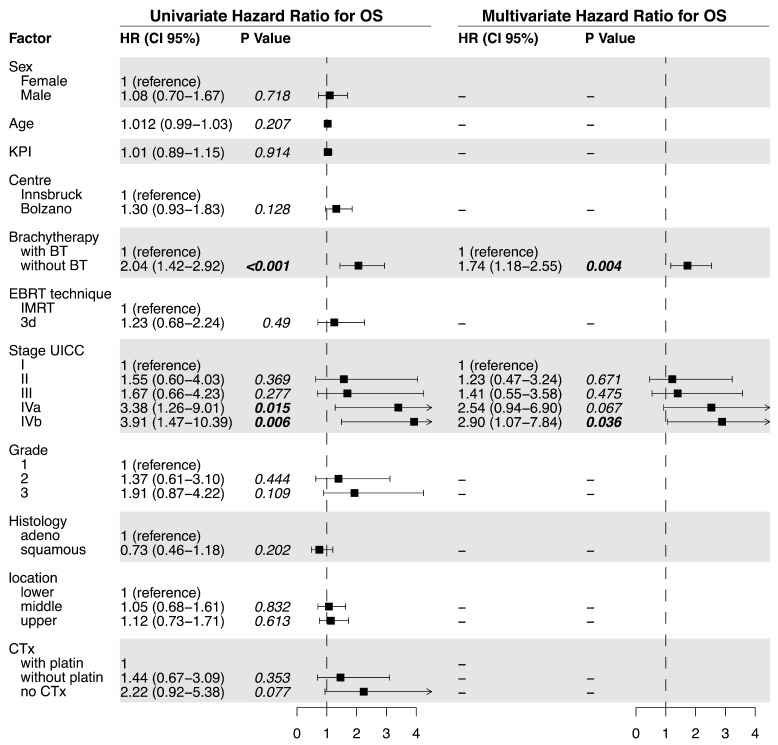
Univariate and multivariate analysis of factors related to overall survival.

**Table 1 cancers-15-03594-t001:** Patient and Tumor Characteristics.

Characteristic	Overall	Without Brachytherapy	With Brachytherapy
**Patients**	183	112	71
**Centre**			
Innsbruck	117 (63.9%)	49 (43.8%)	68 (95.8%)
Bolzano	66 (36.1%)	63 (56.3%)	3 (4.2%)
**Sex**			
female	31 (16.9%)	18 (16.1%)	13 (18.3%)
male	152 (83.1%)	94 (83.9%)	58 (81.7%)
**age**			
median (IQR)	69 (61–76)	69.5 (61–78)	68 (61–75)
**KPS ***			
10	21 (11.5%)	10 (8.9%)	11 (15.5%)
9	53 (29%)	27(24.1%)	26 (36.6%)
8	45 (24.6%)	37 (33.0%)	8 (11.3%)
7	38 (20.8%)	24 (21.4%)	14 (19.7%)
6	12 (6.6%)	4 (3.6%)	8 (11.3%)
5	7 (3.8%)	4 (3.6%)	3 (4.2%)
4	2 (1.1%)	1 (0.9%)	1 (1.4%)
not assessed	5 (2.7%)	5 (4.5%)	0 (0%)
**Histology**			
squamous	143 (78.1%)	91 (81.3%)	52 (73.2%)
adeno	30 (16.4%)	16 (14.3%)	14 (19.7%)
other	10 (5.5%)	5 (4.5%)	5 (7.0%)
**Localization**			
upper third	62 (33.9%)	47 (42.0%)	15 (21.1%)
middle third	66 (36.1%)	33 (29.5%)	33 (46.5%)
lower third	55 (30.1%)	32 (28.6%)	23 (32.4%)
**Stage (UICC) ***			
I	15 (9.4%)	6 (6.3%)	9 (14.1%)
II	28 (17.5%)	11 (11.5%)	17 (26.6%)
III	87 (54.4%)	56 (58.3%)	31 (48.4%)
IV	30 (18.8%)	23 (24.0%)	7 (10.9%)
**EBRT technique**			
3D	162 (88.5%)	101 (90.2%)	61 (85.9%)
IMRT	7 (3.8%)	0 (0%)	7 (9.9%)
VMAT	14 (7.7%)	11 (9.8%)	3 (4.2%)
**Combined chemotherapy ***			
with platinum	13 (7.1%)	11 (9.8%)	2 (2.8%)
without platinum	151 (82.5%)	84 (75.0%)	67 (94.4%)
no chemotherapy	19 (10.4%)	17 (15.2%)	2 (2.8%)

* Chi-squared test *p* < 0.05.

**Table 2 cancers-15-03594-t002:** Univariate analysis of factors related to local control.

	HR (95% CI)	*p*
**Sex**		
Female	1	
Male	2.48 (0.99–6.2)	0.052
**Age (higher vs. lower)**	1.00 (0.97–1.03)	0.997
**KPI (lower vs. higher)**	1.18 (0.96–1.46)	0.109
**Centre**		
Innsbruck	1	
Bolzano	0.94 (0.54–1.63)	0.818
**Brachytherapy**		
with BT	1	
without BT *	2.00 (1.19–3.36)	0.009
**EBRT technique**		
IMRT/VMAT	1	
3D	1.51 (0.61–3.78)	0.374
**Stage UICC**		
I	1	
II	3.48 (0.45–26.66)	0.230
III	3.54 (0.48–26.19)	0.215
IVa *	9.28 (1.18–73.15)	0.035
IVb	6.97 (0.89–54.78)	0.065
**Grade**		
1	1	
2	1.49 (0.5–4.25)	0.487
3	1.07 (0.36–3.20)	0.898
**Histology**		
adeno	1	
squamous	0.69 (0.36–1.3)	0.252
**location**		
lower	1	
middle	1.14 (0.62–2.12)	0.669
upper	1.13 (0.59–2.14)	0.716
**Chemotherapy**		
with platin	1	
without platin	2.55 (0.62–10.47)	0.194
no chemotherapy	1.86 (0.34–10.19)	0.472

* significant values *p* < 0.05

**Table 3 cancers-15-03594-t003:** Accumulative rate of acute toxicities according to CTCAE 5.0.

Acute Toxicities Grade	Without BT	With BT
≥1	104 (92.9%)	69 (97.2%)
≥2	74 (66.1%)	39 (54.9%)
≥3	19 (17.0%)	11 (15.5%)
≥4	3 (2.7%)	1 (1.4%)
5	2 (1.8%)	1 (1.4%)

**Table 4 cancers-15-03594-t004:** Accumulative rate of late toxicities according to CTCAE 5.0.

Late Toxicities Grade	Without BT	With BT
≥1 *	21 (18.8%)	31 (43.7%)
≥2 **	16 (14.3%)	21 (29.6%)
≥3 **	10 (8.9%)	16 (22.5%)
≥4	3 (2.7%)	4 (5.6%)
5	2 (1.8%)	3 (4.2%)

* Chi-squared test *p* < 0.001; ** Chi-squared test *p* < 0.05.

## Data Availability

The data presented in this study are available on request from the corresponding author.

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
