# Peer review of "Chemoradiotherapy Combined with Brachytherapy for the Definitive Treatment of Esophageal Carcinoma"

_cancers, 2023, doi:10.3390/cancers15143594_

Round 1
Reviewer 1 Report
It is my pleasure to review this manuscript entitled “Chemoradiotherapy combined with brachytherapy for the definitive treatment of esophageal carcinoma”. Results of 183 patients with locally limited or locally advanced esophageal cancer treated by definitive CRT with or without brachytherapy were analyzed and reported. The authors concluded that brachytherapy after chemoradiation with single-agent 5- FU represents a safe and effective alternative for dose escalation in the definitive treatment of esophageal cancer.
I have a few concerns about the approach of the authors in dealing with this important question.
Major concerns
· This is not a randomized study and as indicated by the authors, a selection bias might have occurred. By Table 1, KPS (more 9, and 10), stages (more I, and II), and use of combined chemotherapy (less without chemotherapy), are all in favor of case treated with brachytherapy.
· The current standard recommended dose for chemoradiation is 50.4 Gy. It is not clear how patients were selected in this study for the sequential EBRT boost for 9Gy although it was described that this was performed if single agent 5-FU was used as a combined chemotherapy. Clarification is needed.
· It is not clear how patients were selected for brachytherapy. Explanation is needed.
· Reading the manuscript, it is not clear how many patients received sequential EBRT boost in addition to brachytherapy. Did 9 Gy EBRT boost and brachytherapy lead to more treatment related complication?
Minor concerns
· Lines 24, 121, 178, LPFS should be median LPFS.
· Lines 25, 131, 178, Local control should be median local control.
· Lines 105-107, it is not clear how these patients were followed. The follow up schedules (intervals of clinic follow up, CT images etc.) need to be specified.
· Lines 160-161, There were a total of 8 death attributed to treatment related toxicity by adding number of death, Grade 5 acute and late toxicities in Tables 3 and 4.
· Line 183, 5.6% in Table 4.
· Lines 242-246, very poor statement
It is acceptable.
Author Response
Major concerns
- This is not a randomized study and as indicated by the authors, a selection bias might have occurred. By Table 1, KPS (more 9, and 10), stages (more I, and II), and use of combined chemotherapy (less without chemotherapy), are all in favor of case treated with brachytherapy.
To address the issue of an imbalanced distribution of more favorable prognostic factors towards the BT group, including the higher number of patients not receiving chemotherapy in the control group, we have added an additional statistical analysis. This propensity score matched pair analysis finds pairs of patients with best comparable characteristics (including age, KPS, UICC stage, histologic grade, and tumor location). However, already the multivariate analysis included in the prior submission revealed that BT results in improved LPFS, OS and LC independently of these prognostic factors. The now added propensity score matched pair analysis provides an even more accurate assessment of median LPFS and OS times in balanced treatment cohorts. Furthermore, we excluded all no CTX patients from this analysis. In the propensity-score matched population, median LPFS, OS and LC remained significantly improved in the BT groups.
- The current standard recommended dose for chemoradiation is 50.4 Gy. It is not clear how patients were selected in this study for the sequential EBRT boost for 9Gy although it was described that this was performed if single agent 5-FU was used as a combined chemotherapy. Clarification is needed.
We have added additional information to the institutional treatment concept in the methods section of the manuscript (line 100-108)
- It is not clear how patients were selected for brachytherapy. Explanation is needed.
We have clarified selection criteria in the methods section (line 111-116)
- Reading the manuscript, it is not clear how many patients received sequential EBRT boost in addition to brachytherapy. Did 9 Gy EBRT boost and brachytherapy lead to more treatment related complication?
In total, 63 Patients received an EBRT boost and brachytherapy. Regarding the question of potentially more treatment related complications we refer to the newly added supplementary table (Suppl. Table 1 and 2) and added text in the manuscript (Line 214-216). To summarize, no significant difference was recorded between patients receiving either BT or EBRT boost alone and those, which were treated with BT and an EBRT boost (valid for both, acute as well as late toxicity).
Minor concerns
- Lines 24, 121, 178, LPFS should be median LPFS
See corrections in the manuscript
- Lines 25, 131, 178, Local control should be median local control.
See corrections in the manuscript
- Lines 105-107, it is not clear how these patients were followed. The follow up schedules (intervals of clinic follow up, CT images etc.) need to be specified.
See additional specifications in the manuscript (Line 136-138)
- Lines 160-161, There were a total of 8 death attributed to treatment related toxicity by adding number of death, Grade 5 acute and late toxicities in Tables 3 and 4.
Thank you very much for the recognized mistake, which occurred in the reported total count of toxicity related death (erroneous N = 6 vs actually N = 8), as a consequence of the integration of an incomplete Table-section in the manuscript (table A1 / acute toxicities / missing category ‘esophageal perforation’ with two lethal events in the group without BT). The corresponding table has been corrected, and the referring text accordingly adjusted. (Line 201-203 and 214-216)
- Line 183, 5.6% in Table 4.
The incorrect value in the text (4.5%) has been corrected accordingly
- Lines 242-246, very poor statement
We agree with the reviewer’s comment: the mentioned sentence was removed, and replaced by a clarification regarding the available center-specific treatment options, patient selection for the two treatment arms and occurring bias in patient inclusion. (Line 294-313)
Reviewer 2 Report
Although a valuable work, some significant issues need to be discussed to improve the quality of the manuscript:
1. Materials & Methods Chapter:
- The applied technique of brachytherapy is not described (type of applicators); representative figures would help understand the applied method.
- Brachytherapy dose prescription needs include the diameter of the applicator, the reference dose prescription method, and the irradiated tissue depth within the CTV.
- Please state if the patients (especially in higher stages)received endoscopic laser debulking before the treatment and, if any, the timing of the approaches.
2. Discussion chapter:
In the middle section, BT sources are well-centered; therefore, a better result is unsurprising. It is of interest to discuss/demonstrate the fixation(or centralization) method in the upper and lower sections. These difficulties could cause more unfavorable results compared to the middle section locations.
3. Conclusion chapter:
In line 253 authors state: " Contrarily to previous reports ..."; however, two recent large review publications are not listed in the citation list nor discussed in the discussion chapter.
These are:
Rovirosa et al., J Contemporary Brachytherapy 2022; 14, 3:229-309 and Lancellotta et al., Eur Rev Med Pharmacol Sci 2020; 24: 7589-7597
J. Contemporary J Contemp Brachytherapy 2022; 14, 3: 299–309
Author Response
- The applied technique of brachytherapy is not described (type of applicators); representative figures would help understand the applied method.
and
Brachytherapy dose prescription needs include the diameter of the applicator, the reference dose prescription method, and the irradiated tissue depth within the CTV.
Please see the expanded description of the brachytherapy technique in the revised manuscript:
“Intraluminal brachytherapy was performed as 2d planned HDR-afterloading with Ir-192 with 10 Gy prescribed in 5mm mucosal tissue depth in 2 fractions (5 Gy each), 1 fraction per week, and starting two weeks after completion of EBRT (in accordance to the Gec Estro Handbook of Brachytherapy [8]). The maximum applicator surface dose was limited to 11 Gy. The length of the planning target volume included the extent of the tumor plus 2 cm proximal and distal of the lesion. Dose planning was performed in Oncentra Brachy (Elekta Solutions AB, Stockholm, Sweden) using the AAPM TG-43 formalism. The reference dose was calculated at 5 mm mucosal depth. The proximal and distal end of the tumor was marked using radio-opaque surgical clips placed during esophagoscopy a day before the first brachytherapy session. Positioning of the applicator (Bonvoisin-Gérard Esophageal Applicator Set, Nucletron B.V., Veenendaal, NL) was performed using a mobile x-ray device (c-arm) for visualization. A mouthpiece is used for the fixation of positioned applicator. The applicator diameter ranged from 8-12mm (median 10), using the largest possible applicator to achieve a flatter dose gradient and minimize mucosal surface dose.”
- Please state if the patients (especially in higher stages) received endoscopic laser debulking before the treatment and, if any, the timing of the approaches.
Please see added text in the manuscript:
“None of the patient received endoscopic laser debulking prior to brachytherapy. “
- In the middle section, BT sources are well-centered; therefore, a better result is unsurprising. It is of interest to discuss/demonstrate the fixation(or centralization) method in the upper and lower sections. These difficulties could cause more unfavorable results compared to the middle section locations.
As listed in figure 2 and 3, both LPFS and OS were nearly identical irrespective of the location of the tumor. In addition, as shown in figure A2, subgroup analysis of LPFS indicates that tumors located in the middle section least profited from additional brachytherapy in comparison to the lower and upper sections in our cohort. In fact, the hazard ratio for LPFS of BT vs. no BT was 0.36 in the lower, 0.76 in the middle, and 0.34 in the upper section. Therefore, BT is favorable for LPFS in all sections (HR < 1.0), and surprisingly, in the lower and upper section patients seem to profit the most from added BT (however, only according to this subgroup analysis of the primary endpoint).
Since we did not notice any advantage or significant difference in LPFS of middle section over distant section BT, we did no further elaborate on this issue.
- In line 253 authors state: " Contrarily to previous reports ..."; however, two recent large review publications are not listed in the citation list nor discussed in the discussion chapter. These are: Rovirosa et al., J Contemporary Brachytherapy 2022; 14, 3:229-309 and Lancellotta et al., Eur Rev Med Pharmacol Sci 2020; 24: 7589-7597
The respective sentence in the conclusion has been accordingly adapted, and additional literature, including studies mentioned in the two proposed reviews have been cited in the discussion.
Reviewer 3 Report
This is a well-designed study.
However, some issues still are required further addressed and clarified.
1. In the Abstract, Page 1, Line 26-27,
The phrase "the rate acute esophagitis" can be phrasing to "the rate of acute esophagitis."
2. In the Abstract, Page 1, Line 28-29,
In the manuscript, the overall rate of esophageal stenosis was higher in the BT group (22.5 % vs 9.8 %). There was no difference in the occurrence of higher grade (> G3) late toxicities.
However, in the “Table 4. Rate of late toxicities according to CTCAE 5.0”
The late toxicities grade ≥3 was higher in the BT group (22.5 % vs 8.9 %), with Chi-Square p < 0.05, which can be significant difference.
Likewise, the rate of esophageal stenosis was higher in the BT group (grade 1-3 22.5% vs 9.8%; p < 0.05; table A2)
In the Table A2. Highest-grade late toxicities according to CTCAE 5.0.
The late toxicities of esophageal stenosis grade ≥3 was higher in the BT group (9.9% vs 5.4%)
Please carefully verify the values presented in the abstract to ensure their accuracy and consistency.
3. Line 156-157: Likewise, the rate of esophageal stenosis was higher in the BT group (grade 1-3 22.5% vs 9.8%; p < 0.05; table A2)
In the Table A2. Highest-grade late toxicities according to CTCAE 5.0.
The late toxicities of esophageal stenosis grade ≥3 was higher in the BT group (9.9% vs 5.4%)
Please ensure the consistency of the data presented in Table A2.
4. The manuscript contains several typographical errors that should be addressed.
Eg. esophagal in the Line 237, Table A2 (esophagal fistula, esophagal perforation……), which should be corrected as “esophageal”.
5. Line 93, 3-Dimentional or 3D, instead of 3d.
6. Line 111, 116: chi2-test.
Please review the appropriate usage of the chi-squared test or χ2 test in the manuscript.
7. There are still 19(10.4%) patients who did not receive chemotherapy (no CTx)
Please check the inclusion criteria should be patients who receive chemoradiotherapy since the title of this study “Chemoradiotherapy combined with brachytherapy for the definitive treatment of esophageal carcinoma”.
In the Discussion section, it is recommended to provide a more detailed explanation of the "no CTx" group included in this study. Alternatively, considering the potential complexity or ambiguity associated with the "no CTx" group, it might be worth considering excluding this group and subsequently revising the Results section accordingly.
8. Line 121: Local Progression Free Survival (LPFS) has been introduced in Line 61.
Please check the abbreviation LPFS all through the manuscript, particularly in Line 78, 107, 176.
Additionally, it is recommended to check the abbreviation to Local Control (LC) and Overall Survival (OS) to ensure their correctness and consistency as well.
9. In this study, patient with stage IV account for 18.8%. Furthermore, stage IVB significant affect OS and LPFS.
However, in the Line 172-175, the authors investigated the benefit of adding brachytherapy to standard chemoradiotherapy (CRT) in the primary, curative treatment of esophageal cancer.
Please explain and discuss the effect of stage IVB and its interplay with brachytherapy (or brachytherapy is NOT recommended in stage IVB patients?) in the Discussion part.
10. The treatment outcomes may be influenced by various factors, including the treatment volume, experience, and equipment. It is conceivable that there could be differences in Local Progression-Free Survival (LPFS) and Overall Survival (OS) between the two institutions. Therefore, it is advisable to conduct a further analysis of LPFS and OS specifically comparing the outcomes between these two institutions. This analysis will provide valuable insights into the potential impact of institutional factors on treatment efficacy and patient outcomes.
Author Response
- 1 In the Abstract, Page 1, Line 26-27, The phrase "the rate acute esophagitis" can be phrasing to "the rate of acute esophagitis."
See correction in the manuscript
- 2 and 3: In the Abstract, Page 1, Line 28-29, In the manuscript, the overall rate of esophageal stenosis was higher in the BT group (22.5 % vs 9.8 %). There was no difference in the occurrence of higher grade (> G3) late toxicities.
However, in the “Table 4. Rate of late toxicities according to CTCAE 5.0”
The late toxicities grade ≥3 was higher in the BT group (22.5 % vs 8.9 %), with Chi-Square p < 0.05, which can be significant difference.
Likewise, the rate of esophageal stenosis was higher in the BT group (grade 1-3 22.5% vs 9.8%; p < 0.05; table A2)
In the Table A2. Highest-grade late toxicities according to CTCAE 5.0.
The late toxicities of esophageal stenosis grade ≥3 was higher in the BT group (9.9% vs 5.4%)
Please carefully verify the values presented in the abstract to ensure their accuracy and consistency.
Line 156-157: Likewise, the rate of esophageal stenosis was higher in the BT group (grade 1-3 22.5% vs 9.8%; p < 0.05; table A2)
In the Table A2. Highest-grade late toxicities according to CTCAE 5.0.
The late toxicities of esophageal stenosis grade ≥3 was higher in the BT group (9.9% vs 5.4%)
Please ensure the consistency of the data presented in Table A2.
We apologize for the potentially confounding presentation of toxicity data. We have clarified the distinction between cumulative overall toxicities, and the analysis of individual late toxicity items.
In fact, there was no difference in the overall rate of life threatening or lethal (G4 and 5) late toxicities, but there was a significant difference in the cumulative rate of severe (G3 or higher) late toxicities. In the analysis of individual late toxicities items, esophageal stenosis was the only one with significant differences between BT and no BT (grade 1-3 22.5% vs 9.8%; p < 0.05; table A2).
In addition, we recognized that the appendix table A1 was submitted in an incomplete version missing the category of esophageal perforation, thereby causing an inconsistency in the total amount of lethal toxicities (wrong 6 vs correct 8 patients).
Accordingly, substantial modifications and corrections have been performed in the abstract, main text, and tables.
- The manuscript contains several typographical errors that should be addressed. Eg. esophagal in the Line 237, Table A2 (esophagal fistula, esophagal perforation……), which should be corrected as “esophageal”.
See correction in the manuscript
- Line 93, 3-Dimentional or 3D, instead of 3d.
See correction in the manuscript
- Line 111, 116: chi2-test. Please review the appropriate usage of the chi-squared test or χ2 test in the manuscript.
See corrections in the manuscript
- There are still 19(10.4%) patients who did not receive chemotherapy (no CTx)
Please check the inclusion criteria should be patients who receive chemoradiotherapy since the title of this study “Chemoradiotherapy combined with brachytherapy for the definitive treatment of esophageal carcinoma”.
In the Discussion section, it is recommended to provide a more detailed explanation of the "no CTx" group included in this study. Alternatively, considering the potential complexity or ambiguity associated with the "no CTx" group, it might be worth considering excluding this group and subsequently revising the Results section accordingly.
To address the issue of an imbalanced distribution of more favorable prognostic factors towards the BT group, including the higher number of patients not receiving chemotherapy in the control group, we have added an additional statistical analysis. This propensity score matched pair analysis finds pairs of patients with best comparable characteristics (including age, KPS, UICC stage, histologic grade, and tumor location). However, already the multivariate analysis included in the prior submission revealed that BT results in improved LPFS, OS and LC independently of these prognostic factors. The now added propensity score matched pair analysis confirmed the accurate assessment of median LPFS and OS times in balanced treatment cohorts. Furthermore, we excluded all no CTX patients from this analysis. In the propensity-score matched population, median LPFS, OS and LC remained significantly improved in the BT groups.
- Line 121: Local Progression Free Survival (LPFS) has been introduced in Line 61. Please check the abbreviation LPFS all through the manuscript, particularly in Line 78, 107, 176. Additionally, it is recommended to check the abbreviation to Local Control (LC) and Overall Survival (OS) to ensure their correctness and consistency as well.
See corrections in the manuscript
- In this study, patient with stage IV account for 18.8%. Furthermore, stage IVB significant affect OS and LPFS. However, in the Line 172-175, the authors investigated the benefit of adding brachytherapy to standard chemoradiotherapy (CRT) in the primary, curative treatment of esophageal cancer. Please explain and discuss the effect of stage IVB and its interplay with brachytherapy (or brachytherapy is NOT recommended in stage IVB patients?) in the Discussion part.
While the majority of patients were treated in a local or locally advanced stage with curative intent, some patients also received definitive chemoradiation with limited M1 UICC stage IVb, and thus were included in this analysis too. There is a trend to improved LPFS and OS also in this late stage subgroup with brachytherapy (N=7), but of course not in a significant extent. In addition these patients neither developed late toxicities nor were there any acute toxicities larger then grade 2 in this group. However, it is not our intent to recommend chemoradiation plus BT in this setting, and we have clarified this fact in the discussion as advised. Since the stage is included both in the multivariate analysis, and in the matched pair analysis as a potential confounding factor, the inclusion of these patients does not impair the key findings.
- The treatment outcomes may be influenced by various factors, including the treatment volume, experience, and equipment. It is conceivable that there could be differences in Local Progression-Free Survival (LPFS) and Overall Survival (OS) between the two institutions. Therefore, it is advisable to conduct a further analysis of LPFS and OS specifically comparing the outcomes between these two institutions. This analysis will provide valuable insights into the potential impact of institutional factors on treatment efficacy and patient outcomes.
The centers of this trial share identical treatment concepts since the center in Bolzano is an international spin-off of the center in Innsbruck, and the physicians of Bolzano received their specialty training in Innsbruck. Except for the unavailability of brachytherapy in the Italian center, both centers are comparably equipped. However, multivariate analysis included the center as a potential confounding factor, and a center specific difference in LPFS and OS could not be detected (see figures 2 and 3). Accordingly, we have added information also in the discussion section of the manuscript (line 294-302)
Round 2
Reviewer 2 Report
All reviewer issues were adequately answered - the manuscript quality was improved.
Author Response
We thank the reviewer for the previous comments and valuable suggestions
Reviewer 3 Report
However, there are still concerns regarding the title of Table 2 and Figure 2 that require correction and a response.
Minor concern:
1.In the manuscript LINE 198: Table 2. R Univariate analysis of factors related to Local Control
The "R" in the title of Table 2 is unnecessary and should be removed.
2.
Figure 2 is missing in the revised version of the manuscript.
The previously mentioned issue remains unanswered.
Please verify the completeness of Figure 2 and provide a response to the following issue once again.
The treatment outcomes may be influenced by various factors, including the treatment volume, experience, and equipment. It is conceivable that there could be differences in Local Progression-Free Survival (LPFS) and Overall Survival (OS) between the two institutions. Therefore, it is advisable to conduct a further analysis of LPFS and OS specifically comparing the outcomes between these two institutions. This analysis will provide valuable insights into the potential impact of institutional factors on treatment efficacy and patient outcomes.
Author Response
- Minor concern: In the manuscript LINE 198: Table 2. R Univariate analysis of factors related to Local Control - The "R" in the title of Table 2 is unnecessary and should be removed.
The mistake has been corrected
- Figure 2 is missing in the revised version of the manuscript. The previously mentioned issue remains unanswered.
Please verify the completeness of Figure 2 and provide a response to the following issue once again.
The treatment outcomes may be influenced by various factors, including the treatment volume, experience, and equipment. It is conceivable that there could be differences in Local Progression-Free Survival (LPFS) and Overall Survival (OS) between the two institutions. Therefore, it is advisable to conduct a further analysis of LPFS and OS specifically comparing the outcomes between these two institutions. This analysis will provide valuable insights into the potential impact of institutional factors on treatment efficacy and patient outcomes.
We apologize for not having answered to this previous concern to a sufficient extent. In fact, figures 2 and 3 do not include the center in the “multivariate” part, however this is on purpose: We have performed a univariate and multivariate analysis of factors potentially influencing LPFS and OS. The center (Bolzano vs Innsbruck) was included in this analysis. As described in the now expanded statistical methods, we performed a stepwise backward elimination of non-significant factors for the multivariate analysis. In the final model, the center was thus not included. For this reason the center is not listed in the multivariate column of figures 2 and 3, and we can confirm completeness of the two figures as presented in the original manuscript.
However, in order to clearly address the topic of any center specific difference in OS and LPFS due to the factors mentioned by the reviewer (treatment volume, experience, and equipment), we have added another analysis. This included comparable patients from the two centers who have not received brachytherapy (as BT was only performed in one center). The results confirmed the previously stated fact that there is no statistically significant difference in center specific OS and LPFS (see addition in the results of the manuscript). We have now also added two figures showing Kaplan-Meier plots of center specific OS and LPFS to the supplementary materials. We hope that the performed modifications and additions sufficiently address the reviewers concern.